# Biochemical and Metabolic Plant Responses toward Polycyclic Aromatic Hydrocarbons and Heavy Metals Present in Atmospheric Pollution

**DOI:** 10.3390/plants10112305

**Published:** 2021-10-26

**Authors:** Lázaro Molina, Ana Segura

**Affiliations:** Department of Environmental Protection, Estación Experimental del Zaidín, C.S.I.C., Calle Profesor Albareda 1, 18008 Granada, Spain; ana.segura@eez.csic.es

**Keywords:** atmospheric pollutants, polycyclic aromatic hydrocarbons, heavy metals, ROS production, phytohormones, P450 monooxygenases, glutathione, plant responses

## Abstract

Heavy metals (HMs) and polycyclic aromatic hydrocarbons (PAHs) are toxic components of atmospheric particles. These pollutants induce a wide variety of responses in plants, leading to tolerance or toxicity. Their effects on plants depend on many different environmental conditions, not only the type and concentration of contaminant, temperature or soil pH, but also on the physiological or genetic status of the plant. The main detoxification process in plants is the accumulation of the contaminant in vacuoles or cell walls. PAHs are normally transformed by enzymatic plant machinery prior to conjugation and immobilization; heavy metals are frequently chelated by some molecules, with glutathione, phytochelatins and metallothioneins being the main players in heavy metal detoxification. Besides these detoxification mechanisms, the presence of contaminants leads to the production of the reactive oxygen species (ROS) and the dynamic of ROS production and detoxification renders different outcomes in different scenarios, from cellular death to the induction of stress resistances. ROS responses have been extensively studied; the complexity of the ROS response and the subsequent cascade of effects on phytohormones and metabolic changes, which depend on local concentrations in different organelles and on the lifetime of each ROS species, allow the plant to modulate its responses to different environmental clues. Basic knowledge of plant responses toward pollutants is key to improving phytoremediation technologies.

## 1. Introduction

Atmospheric pollutants are considered compounds that are not normally present in air and are present at higher concentrations than usual or that are abnormally present in certain atmospheric layers [1]. Air pollution’s composition is complex and variable, depending on time, geographic zone, climate conditions, human activities and many other factors [2].

Air pollution is mainly formed by a gaseous fraction and by particulate matter [3]. Gases such as mono- and dioxide (CO, CO_2_), methane (CH_4_) and volatile organic compounds (VOCs), represent the main carbonaceous compounds of air contamination. Nitrogen, mainly in the form of ammonia (NH_3_), ammonium (NH_4_^+^), dinitrogen tetroxide (N_2_O_4_), nitrogen mono- and dioxide (NO, NO_2_), sulphur dioxide (SO_2_), ozone (O_3_), mercuric vapors (Hg), chlorine (Cl_2_) and fluorides (HF, SiF_6_, CF4 and F_2_) are other gases present in air pollution. Most of these compounds are very reactive, and interactions between them and with other atmospheric molecules can form other, even more harmful compounds. Amongst the VOCs, solvents such as benzene, toluene, ethylbenzene, and xylenes (BTEX), hexane (C_6_H_6_) and phenol vapours are the main organic carbon forms found in gaseous fractions of atmospheric pollution.

Atmospheric particulate matter (PM) or atmospheric aerosols are mostly constituted by inorganic ions, such as nitrate, sulphate and ammonium and mineral dust, sea salt and carbon derivatives, i.e., black carbon. PAHs represent ~1.25% of atmospheric particulate matter and can be found at concentrations in the range of ng m^−3^ [4]. Heavy-weight PAHs [HMW-PAHs] (more than four aromatic rings) are the most abundant PAHs in PM [5]. Heavy metals (HMs) such as arsenic (As), cadmium (Cd), chromium (Cr) and lead (Pb) have also been found at the same average concentrations as PAHs (~1.37%). Pb and As seem to be the most abundant HMs found in atmospheric PM [6]. Although trichloroethylene (TCE), polychlorinated biphenyls (PCBs), BTEX (benzene, toluene, ethylbenzene and xylenes), dioxins and others are also toxic constituents of atmospheric contamination, HMs and PAHs are among the most abundant compounds in PM and they constitute the main focus of this review.

Atmospheric PAHs are produced as the consequence of pyrolysis and the incomplete combustion in deficient oxygen conditions, not only of solid fuels, waste and plant residues, but also as a consequence of volcanic eruptions or natural fires [7,8]. Heavy metals are chemical elements naturally found on Earth. However, human activities have altered their biogeochemical cycles, allowing their accumulation at concentrations and/or locations where they exert a deleterious effect on organisms. Nevertheless, some natural processes can also result in local accumulations of HMs; i.e., high concentrations of As in water are mainly the result of rock-weathering, volcanic eruptions or microbial activity. The most significant sources of HM emissions to the atmosphere are smelters (iron, steel and non-ferrous metallurgy), fossil fuel combustion and mining activities, though, there are many others, depending on the HM [9].

It is estimated that the impact on health problems related with air pollution will increase worldwide, reaching 3.1 million premature deaths annually by 2030 (OECD Environment Outlook to 2030). The impact of PAHs and HMs on human and animal health has been widely studied. PAHs (especially those of high molecular weight [HMW-PAHs]) act as potent carcinogens; they can affect the immune, reproductive, hematopoietic and nervous systems [10]. Although some HMs are necessary for the correct functioning of certain enzymes in humans, excessive amounts of some HMs, such as nickel (Ni), copper (Cu) and zinc (Zn), are potentially toxic. Others have adverse effects on human health even at very low concentrations (i.e., Pb and Cr) [11].

Plants are exposed to these toxic compounds not only through their aerial parts but also in below-ground organs because of the deposition of PM in soils. In fact, the amount of PAHs in soil is high, not only in many industrial areas, but also in non-industrial soil [12]. In this review we summarize the effects of PAHs and HMs of atmospheric contaminants on plants and the defensive responses that are triggered in plants in response to them.

## 2. PAHs and HMs Affect Seed Germination and Plant Growth

PAHs and some of their byproducts, formed during the natural processes of PAH modification by ageing, biodegradation and weathering, affect the rate of seed germination and seedling weight [13,14]. For example, some photo-induced PAHs are more toxic than their parental compounds, probably because they have higher water solubility [13,15]. In fact, seed germination has often been used as a physiological index test to examine the toxic effects of a certain contaminant on plants. However, the effects observed depend not only on the plant species but also on; (i) the PAH type; (ii) PAH local concentrations; (iii) PAH solubility in water (generally correlated with PAH bioavailability), (iv) organic matter content and soil texture and (iv) the age of the contamination [16,17]. Therefore, low-molecular-weight PAHs (LMW-PAHs), which have higher water solubility and bioavailability than HMW-PAHs, are generally more toxic to plants than HMW-PAHs [13]; different types of soil, of low organic matter and grainy texture, retain less PAHs and therefore plant germination is improved when compared with compact soils of high organic matter content [16,18,19] and the phytotoxicity of PAH mixtures is higher at the early stages of contamination than in aged contaminated soil because of the loss of volatile compounds (mainly LMW hydrocarbons) with time and the adsorption of PAHs into organic matter and colloids in the soil (with the concomitant reduction of bioavailability) [20,21,22].

As reported in many other organisms, hormesis has been identified in plant responses toward different stressful agents, among them PAHs and HMs [23]. Hormesis is defined as “an adaptive response of biphasic dose where it responds to a stress determining factor, in which sub-doses induce stimulation and high doses induce inhibition” [24]. In plants, the induction of hormesis leads to responses that optimize many physiological processes (i.e., increases in chlorophyll content, alteration of signalling pathways, and others) which, in turn, enhance seed germination, crop growth and early flowering [25,26]. As many of the cellular responses toward pollutants converge at some point with responses toward other compounds, i.e., plant pathogens, hormesis has also been associated with cross-resistance toward different stresses [26].

However, the presence of PAHs or HMs above certain doses has detrimental effects on plant germination and growth and biomass yield [27,28,29]. Toxic amounts of PAHs lead to shorter roots and lower root weight in seedlings than develop in the absence of the contaminant [30,31]. Pollutants cause a mechanical disruption of cellular membranes, diminishing their capacity to retain water and nutrient uptake and alteration of cell expansion processes due to disruption of the cell organelle’s metabolism and the alteration of hormone actions (auxins) [30,31]. Other effects of the presence of contaminants involve a significant reduction in cell size and mitotic activity [32], and slower expansion of cotyledons following emergence [33]. Furthermore, PAHs produce an inhibition of the growth and chlorophyll content of the seedlings. Many of these effects are due to oxidative damage suffered in the presence of the contaminant [34].

The toxic effects of PAHs depend not only on the physicochemical properties of the contaminant or intrinsic tolerance of the plant, but also on the capacity of natural microbial populations to degrade PAHs and the capacity of the plant to stimulate indigenous soil microbes to degrade contaminants [35,36]. The ability of the plant to stimulate the beneficial capacities of their associated microbiota depends on the composition of the root exudate, chemical properties of the contaminant, soil properties and environmental conditions [37,38].

The presence of HMs in soil also has negative consequences for plants and include overall morphological abnormalities, reductions in dry weight, decrease in germination, and reduced root and shoot elongation [29]. The observed reduction in germination is a consequence of oxidative damage causing membrane alterations, alterations of sugar and protein metabolism, nutrient loss and reduced amounts of total soluble protein levels [39]. The inhibition of many enzymes involved in the digestion and mobilization of food reserves during germination, such as amylases, proteases and ribonucleases, has been reported as one of the effects of HM toxicity [39,40,41,42]. The toxic effect of HMs on seeds depends on the particular heavy metal affecting them; in *Arabidopsis thaliana* seeds, the reported decrease in seed germination from contamination followed the order of Hg^2+^ > Cd^+^ > Pb^2+^ > Cu^2+^ [29]. HMs can also be oxidized or become complex entities in soil, sometimes increasing their toxicity [43].

It has been proposed that HMs exert toxicity in plants through four possible mechanisms: (i) similarities with the nutrient cations (for example, it has been reported that As and Cd compete with P and Zn, respectively, for their absorption); (ii) the direct interaction of HMs with sulfhydryl groups (-SH) of functional proteins, which disrupt their structure and provokes its inactivation; (iii) the inactivation of proteins by the displacement of essential cations from specific binding sites and (iv) the generation of reactive oxygen species (ROS), which subsequently damage essential macromolecules [44].

## 3. PAHs and HMs Affect Plant Metabolism

The toxicity of PAHs and HMs affects plant metabolism in different aspects. By using –omics’ techniques, some of the most important effects of these contaminants in plant physiology are being revealed (Figure 1).

### 3.1. Effects on the Photosynthetic System

The presence of PAHs results in a reduction in total chlorophyll content of both C3 and C4 plants, with an increase of the chlorophyll a/b ratio, which is one of the direct indications that the plants are experiencing extremely harmful conditions [45]. PAHs inhibit RuBisCO carboxylation activity, decreasing photosynthetic rates and inhibiting photosystem II activity, blocking the photosynthetic electron flow from photosystem II to photosystem I (Figure 1). This restriction of the electrons flux is primarily due to the net degradation of the D1 protein, which is caused by the accumulation of (ROS) in PAH-treated plants [46,47]. As mentioned above, PAHs probably alter membrane permeability with subsequent production of ROS, which produces this functional change in PSII [48].

Similar effects have been described in plants under HM stress. Disruption of the photosynthetic machinery by HM stress is inferred from the low abundance of proteins involved in the Calvin cycle and the photosynthetic electron transport chain and by the drastic reduction in abundance/fragmentation of large and small sub-units of RuBisCO (LSU and SSU) [49].

Interestingly, mild concentrations of these ions (1 μM) produce an increase of proteins involved in photosystems I and II and the Calvin cycle (Figure 1). This effect might be an adaptive strategy for overcoming plant injury; the presence of high quantities of photosynthetic assimilated into respiration would help plants to yield more energy, needed to combat heavy metal stress [50].

### 3.2. Effects on Carbon Metabolism

A combination of metabolomic, proteomic and transcriptomic studies have determined that the application of phenanthrene on wheat leaves affects the functioning of the tricarboxylic acid cycle (TCA) [51]. The presence of this PAH produces alterations in the concentrations of the TCA intermediates, increasing citrate and malate and decreasing α-ketoglutarate, fumarate, oxaloacetate, pyruvate and succinate (Figure 1). The accumulation of citrate and malate is due to the induction of the expression of the pyruvate dehydrogenase, dihydrolipoyllysine-residue succinyltransferase, fumarate hydratase and ketoglutarate dehydrogenase and the inhibition of NADH synthesis, isocitrate dehydrogenase and malate dehydrogenase, GTP formation, succinyl-CoA synthase and the respiratory chain linked to the succinate reductase. Ultimately, the altered functioning of the TCA cycle was due to a decrease in the cellular pyruvate concentrations under exposure to phenanthrene, an observation also reported in the root cells of wheat plants [52]. Other important metabolic enzymes that have been shown to be down-regulated in the presence of phenanthrene in wheat are glyceraldehyde-3-phosphate dehydrogenase (NADH-forming enzyme) and the adenosine kinase, involved in the synthesis of ATP [53]. Similarly, stress caused by HMs also produces changes in the concentrations of TCA intermediates (mainly, decreasing malate or succinate) and changes in the expression of enzymes of this important metabolic pathway, indicating important disturbances that affect the energy potential (the synthesis of NADH and GTP and the correct functioning of the respiratory chain) of the plant cell [54,55] (Figure 1). For both stresses, the reported results suggest that there is an inhibition of the energy-forming processes, i.e., the synthesis of ATP and NADPH, and an activation of a fermentative metabolism in plants cells.

Galactose, sucrose, inositol galactoside and melibiose metabolisms are activated in the presence of PAHs, increasing the content of the D-mannose, D-galactose, raffinose, galactinol, melibiose, sucrose, and D-glucose metabolites in plant tissues. As mentioned above, PAHs cause decreases in water content and in the nutrient-utilization efficiency in plants, besides provoking the inhibition of photosynthetic activity and electron transport (Figure 1). Therefore, the accumulation of sugars (or derivatives) can supply compounds for the higher demand of energy required to tolerate adverse stress conditions created by the presence of PAHs. Furthermore, they can also act as osmolytes in the protection of cellular structures and to sustain osmotic balance under the water stress conditions caused by the PAHs [56,57]. In the presence of PAHs, a decrease in the content of hydroxypyruvate and the metabolism of certain amino acids has been observed, affecting the gluconeogenic pathway [56].

Plants exposed to HMS also suffer alterations in the metabolism of glucose (Figure 1). A decrease in the glycolytic flux due to a dysfunction of the pyruvate kinase glyceraldehyde-3-phosphate dehydrogenase or of enolase, key regulators of glycolysis, has been observed in presence of Hg^2+^, As or Cu heavy metals, respectively [58,59]. In these studies, to compensate for the cellular deficit of NADH, the anaplerotic NADP-dependent malic enzyme was induced; this induction was correlated with a decrease in malate content observed in heavy metal-exposed plants. This enzyme produces cytosolic pyruvate to also compensate for pyruvate kinase dysfunction. This malic enzyme has also been suggested to be involved in plant defence responses against oxidative damage and to compensate for the large energy requirements of NADPH and to provide pyruvate to supply the mitochondrial TCA cycle [54].

### 3.3. Effect on Amino-Acid and Nitrogen Metabolism

Plants exposed to PAHs significantly increase the activity of nitrate reductase (Figure 1), an enzyme responsible for nitrate assimilation by plants through the reduction of nitrate to ammonia [52]. This increase in ammonia content, and the upregulation of the glutamate dehydrogenase (GDH2), may explain the augmentation of the levels of glutamic acid [57] and the decrease of α-ketoglutarate levels [51] observed under PAH stress. Proline, which forms part of the non-enzymatic responses to ROS, is formed from glutamate, and it has been reported to accumulate under both stresses [56,57]. HMs also increase nitrogen metabolism by enhancing protease activity [60], inhibiting the synthesis of proteins [55] and reducing the activity of nitrate and nitrite reductases, enzymes involved in nitrate assimilation, and of glutamine synthetase, glutamine oxoglutarate aminotransferase and glutamate dehydrogenase, which are involved in ammonia assimilation [61,62].

The synthesis of branched chain amino acids (L-valine, L-leucine and l-isoleucine) from pyruvate have been shown significantly upregulated under PAH and HM exposure [55,56,57,63] (Figure 1), and this increased utilization of pyruvate may explain the lower content of this compound found in PAH-stressed plants by Zhan and co-workers [52]. The presence of PAHs has also produced an increase in L-alanine, L-tryptophan, L-(−)-tyrosine and D-(+)-phenylalanine content in plant cell tissues [56,57]. The three latter amino acids are precursors in important pathways for the biosynthesis of secondary metabolites [64].

In the presence of phenanthrene enhanced sulphur assimilation from sulphite, serine transformation and increased cysteine synthesis has been demonstrated [57] (Figure 1). Cystein is a powerful antioxidant and facilitates nitrate absorption and/or foliar transport [65]. Furthermore, cysteine, glutamate and glycine are the precursors of the antioxidant glutathione [66]. The expression of the enzymes involved in the glutathione cycle—glutamate cysteine ligase (synthesis), glutathione reductase (recycling) and glutathione-S-transferase (the transfer to xenobiotics)—is upregulated in the presence of PAHs [57].

Aminoacyl-tRNA biosynthesis, involved in the biosynthesis of proteins, is also significantly increased by plants’ exposure to PAHs. This could be due to the increase in demand for anti-oxidative enzymes, stress proteins and DNA repair enzymes [57].

### 3.4. Effects on Secondary Metabolism

Many studies have shown elevations in the content of plant polyphenols (Figure 1), which play an important role in antioxidant plant responses, in response to both PAH and heavy-metal treatments. This is possible a consequence of the stimulation of phenylalanine ammonia-lyase (PAL) activity, and also of the increase in the concentration of the precursors phenylalanine, tyrosine and tryptophan. Tyrosine is first converted to 4-hydroxyphenylpyruvate, which is subsequently transformed to turinic acid by the action of the 4-hydroxyphenylpyruvate dioxygenase. Turinic acid is a precursor of tocopherols, such as vitamin E and plastoquinone, and improves plant stress resistance. Tryptophan is a precursor of numerous secondary metabolites, such as auxins, antitoxins, glucosinolates and alkaloids that augment aromatic compound biosynthesis [57]. Phenylalanine participates in the biosynthesis of several phytochemicals and antioxidants in the phenylpropanoid pathway [64].

Some authors have indicated that there is a significantly positive dosage relationship between polyphenolic metabolism intensity and contamination levels [67]. However, a significant reduction of phenolic compounds (flavonoids, anthocyanins, tannins, lignins, phenolic acid and the related compounds coumarin, flavenol, cinnamic acid, cinnamic alcohol, cinnamic aldehyde), greater than >40%, was observed in plants exposed to high concentrations of PAH/HM pollution when compared with non-exposed cells [68,69]. It has been hypothesized that when plants cannot counteract oxidative stress, the plant enters into a state of metabolic distress, compromising its secondary metabolism.

Membrane lipid peroxidation has been shown in response to PAH stress [5] where, therefore, it increased the content of several lipids, such as 13-hydroperoxy-9, 11-octadecadienoic acid (13-HPODE), 9-hydroxy-(10E,12Z,15Z)-octadecatrienoic acid, 14,15-dehydrocrepenynic acid, palmitaldehyde, octadeca-11E,13E,15Z-trienoic acid and α-linolenic acid, which have been observed in plants exposed to PAHs.

## 4. Adsorption, Absorption and Accumulation of PAHs and HMs by Plants

### 4.1. Adsorption

Atmospheric PM containing PAHs and HMs can be deposited directly onto plant leaves or in soil. The retention of PMs on leaves depends on the PM atmospheric concentration [70,71], the exposed surface area and leaf-surface properties and topography, which are conditioned by leaves’ hairiness or cuticle compositions [72,73,74,75]. For example, the gymnosperm *Pinus silvestris* can accumulate up to 19 micrograms of PAHs per gram of dry weight of needles [76] and is one of the plant species with the highest levels of PAH accumulation described in the literature; the waxy surface of the pine needles traps PM and gaseous pollutants [77].

Besides being directly deposited on leaves or soil, PMs can also be mobilized from soil to leaves by wind or evaporation, be transported from roots to leaves or be deposited on soil through plant biomass decay (Figure 2; [78,79,80,81]).

### 4.2. Absorption

The uptake of atmospheric contaminants by plant roots varies significantly, depending on factors such as pollutant concentrations in soil, the hydrophobicity of the contaminant, plant species and tissue and soil microbial populations [72,82]; it also depends on temperature [83].

The absorption of LMW-PAHs to the inner tissues of the leaf is mainly conducted by passive diffusion through the hydrophobic cuticle and the stomata. HMW-PAHs are mostly retained in the cuticle tissue and its transfer to inner plant components is limited by the diameters of its cuticle pores and ostioles [84].

PAHs, adsorbed on the lipophilic constituents of the root (i.e., suberine), can be absorbed by root cells and subsequently transferred to its aerial parts [85]. Once inside the plant, PAHs are transferred and distributed between plant tissues and cells in a process driven by transpiration. A PAH concentration gradient across plant–cell components is established, and PAHs are accumulated in plant tissues depending on their hydrophobicities [86]. Almost 40% of the water-soluble PAH fraction seems to be transported into plant roots by a carrier-mediated and energy-consuming influx process (a H^+^/phenanthrene symporter and aqua/glyceroporin) [87,88]. The PAH distribution pattern in plant tissues and in soil suggests that root uptake is the main entrance pathway for HMW-PAHs. Contrarily, LMW-PAHs are probably taken-up from the atmosphere through leaves as well as by roots [89].

Although HM absorption by leaves was first reported almost three centuries ago [90], the mechanism of absorption is not yet fully understood [91]. Absorption mainly occurs through stomata, trichomes, cuticular cracks, lenticels, ectodesmata and aqueous pores [92], with the stomata and trichomes being the preferential sites of ion penetration due to the existence of polar domains in these structures [93]. Transportation to other plant tissues occurs via the phloem vascular system, by mechanisms similar to those transporting photosynthates within the plant. This active HM transport depends on plant metabolism and varies with the chemistry of the HMs. Immobile metals, i.e., Pb, may precipitate or bind to ionogenic sites located on the cell walls, avoiding their movement within the plant leaves. However, these immobile metals can also be transported inside plants under other conditions; i.e., if the levels of HMs are low enough not to surpass their solubility limits, “immobile” metals can move within plants with other metabolites. Alternatively, “immobile” metals may form chelates or complexes with organic compounds present in the phloem. These compounds inhibit metals’ precipitation and favour their transport [91].

However, the soil-root transfer of metals seems to be the major HM entrance pathway [94]. The uptake of HMs by roots mainly depends on the metal’s mobility and availability; that is, in general, it is controlled by soil adsorption and desorption characteristics [95,96]. The key influencing factors inolved include pH, soil organic matter, cation exchange capacity, oxidation-reduction status and the contents of clay minerals [97,98]. At a low pH, the transfer of HM into soils is generally accelerated, while greater organic matter content depletes oxygen and increases the resistance of soil to weathering, preventing heavy metal dissolution [99]. After adsorption into root surfaces, metals bind to polysaccharides of the rhizodermal cell surface or to carboxyl groups of mucilage uronic acid. HMs enter the roots passively and diffuse to the translocating water streams [100]. Metal transportation from roots to the aerial parts occurs through the xylem system, transported as complex entities with different chelates, and is generally driven by transpiration [91].

### 4.3. Accumulation

Several groups of plants have developed the capacity to hyperaccumulate contaminants. Several species of the Poaceae and Fabaceae families, e.g., white clover (*Trifolium repens*), a few vegetable crops, such as carrot (*Daucus carota*), celery (*Apium graveolens*), barley (*Hordeum vulgare*), cabbage (*Brassica oleracea*), soybean (*Glycine max* L.) and spinach (*Spinacia oleracea*), mosses and both broadleaf and conifer trees have been considered as effective PAH accumulators [101,102]. Two mechanisms have been described for the hyperaccumulation of PAHs; one is the production of high quantities of low-molecular-weight organic acids in the root exudates. These acids promote the availability of PAHs by disruption of the complexes in the PAH–soil matrix [103]. PAH-hyperaccumulating plants present higher lipid (membrane and storage lipids, resins, and essential oils) and water content, lower carbohydrate content and a higher plant transpiration-stream flow rate than non-accumulating plants [104]. An additional mechanism for the higher uptake of PAHs in these hyperaccumulating plants is the presence of oil channels within the roots and shoots in plants such as carrots, and high lignin and suberin content that may also absorb organic chemicals [104,105].

Metallophytes are plants that are specifically adapted to soil enriched in HMs [106]. Some metallophytes are hyperaccumulators; they can accumulate 100–1000-fold higher shoot metal concentrations (without yield reduction) compared with non-accumulator plants [107]. They can tolerate the presence of 100 mg kg^−1^, in dried foliage, of Cd, Se or Ti; 300 mg kg^−1^ of Co, Cu or Cr; 1000 mg kg^−1^ of Ni, Pb or As; 3000 mg kg^−1^ of Zn; 10,000 mg kg^−1^ of Mn without showing any visible phenotypical changes [106,108,109]. Many of these plants belong to the Brassicaceae, Phyllanthaceae, Asteraceae or Laminaceae families [107], The biological significance of this phenotype, besides survival in heavily contaminated sites, is that metal hyperaccumulation in leaves could be a defensive mechanism against herbivores (by making leaves unpalatable or toxic) and pathogens [110]. This process requires increased metal uptake and xylem loading, as well as enhanced metal accumulation by sequestration in the apoplasts or vacuoles and detoxification in shoots [111].

## 5. Detoxification of PAHs and HMs by Plants

Plants can detoxify contaminants, mainly by immobilization in cellular compartments such as vacuoles or cell walls. However, some leguminous plants, such as alfalfa (*Medicago sativa* L.) and sorghum (*Sorghum bicolor*), can exude enzymes, such as tyrosinases, laccases or peroxidases, through their roots. These secreted enzymes play an important role in the polymerization reactions that lead to pollutant immobilisation in humic acids in soil, rendering pollutants biologically inaccessible [112]. Furthermore, these enzymes catalyse the oxidation of phenolic compounds and PAHs using hydrogen peroxide as the electron acceptor, transforming these molecules into more easily degradable compounds for the indigenous microbiota, and therefore, indirectly, detoxifying these environments. Similarly, root exudates of various plant species, such as fescue grass (*Festuca arundinacea*), switch grass (*Panicum virgatum*), maize (*Zea mays* L.), soybean, sorghum, alfalfa and clover, have the ability to enhance PAH biodegradation, probably because plant roots can stimulate soil microbial biomass and oxygen transport to the rhizosphere, thus facilitating the degradation process [113,114].

However, once a contaminant is within a plant’s cells, immobilization is the main detoxification pathway. The immobilization pathways are different for organic compounds (such as PAHs) than for HMs (Figure 3).

### 5.1. Detoxification of Organic Compounds

Organic compounds are firstly modified by the action, mainly, of cytochrome P450 monooxygenases [115]. CYP450s are heme-thiolate monooxygenases that use electrons from NADPH to activate molecular oxygen and to insert a single oxygen atom into their substrates. They usually catalyse the hydroxylation or epoxidation, the dealkylation of methoxy or amine substituents and the reductive dehalogenation of aromatic rings, but catalysing the opening of aromatic rings has never been reported [116]. Under normal conditions, CYP450s are involved in the metabolism of a wide variety of natural compounds, such as hormones, lipids and secondary metabolites. Recently, transcriptomic assays have revealed the importance of some dioxygenases, enzymes that are able to oxidize aromatic compounds by the incorporation of two hydroxyl groups, in the first step of the PAH modification in *A. thaliana* plants exposed to phenanthrene [117]. In addition, other oxidoreductases (including peroxidases) and carboxylesterases have been implicated in PAH dissipation [118]. These reactions transform the contaminants into less hydrophobic compounds and increase their reactivity, in what is known as phase I of detoxification (Figure 3) [119].

Phase II involves the conjugation of contaminants with glutathione, amino acids, proteins, peptides, organic acids, mono- and oligo-polysaccharides, lignin and others, resulting in the formation of peptide-, ether-, ester- or thioether-conjugates and the production of hydrophilic compounds [119]. These conjugation reactions are catalysed by different transferases: glutathione S-transferase, glucuronosyl-O-transferase, malonyl-O-transferase, glycosyl-O-transferase, N-glycosyl-transferase, N-malonyl-transferase and others [120]. The main transferases involved in this phase II are glutathione S-transferases (GSTs) and glycosyl-transferases (Figure 3). GSTs represent a family of more than 25 different enzymes that bind glutathione (g-Glu–Cys–Gly) to reactive molecules, protecting the cell from oxidative damage. Glycosyl-transferases have an important role in sugar metabolism and in plant secondary metabolism under normal conditions, and, in the presence contaminants, participate in plant defence and stress tolerance [121]. The formation of these conjugates is a key process in the detoxification of contaminants in plants [101]; conjugates can be kept inside a cell for a certain period without any visible pathological symptoms, mainly because their toxicity is decreased, compared with that of the parental compounds [120]. In some cases, more than 70% of absorbed organic pollutants in plants are accumulated in the form of conjugates [122].

However, as, in most cases, plants do not possess excretion systems, the final destination of the conjugates or the hydroxylated contaminants is their storage in defined compartments of the plant such as cell walls and vacuoles [117,123]. This phase of the process (phase III; Figure 3) allows plants to eliminate pollutants from the vital parts of cells [119,120,121,124]. Conjugates are actively transported to the vacuole and, in some cases, to the apoplast by the action of an ATP-dependent membrane pump [125,126,127]. Dihydroxylated pollutants can also be covalently linked with plant cell-wall polymers and lignin [128,129], probably through the action of cell-wall- or vacuole-associated enzymes (i.e., internal peroxidases and laccases). These enzymes, normally involved in the detoxification of H_2_O_2_, have been also associated with the formation of tyrosine or ferulate cross-links between different plant cell wall polymers with the non-specific oxidative polymerization of phenolic units to produce lignin and with the deposition of aromatic residues of suberin on the cell wall [130].

Therefore, within the plant, PAHs are frequently found as: (i) residues covalently bound to the plant cell wall components (lignin, hemicellulose, cellulose and proteins); (ii) as glutathionylated and glucosylated derivatives located in vacuoles or (iii) mono- or dihydroxylated PAHs or metabolites in plant cells [131].

Recent studies have determined that organic compound sequestration, metabolization and/or dissipation from PAHs takes place mostly in specialized plant tissues or structures such as trichomes, shoot hairs derived from the epidermal cell layer, pavement cells or stomata, in *A. thaliana*, alfalfa, or *Thellungiella salsuginea,* and in the basal salt gland cells on the *Spartina* species [132,133,134,135].

### 5.2. Detoxification of HMs

Plants have developed different mechanisms for HM detoxification. One of them is the excretion of HMs from plant cells by different types of transporters (aquaporins, efflux pumps and others) (Figure 3). HMs can also be chelated by low-molecular-weight molecules such as glutathione, phytochelatins or metallothioneins that facilitate the transport of metals to vacuoles (Figure 3). Glutathione plays an important role in the cellular redox balance and can bind to several metals and metalloids [136]. The two best-characterized heavy metal-binding ligands in plant cells are the phytochelatins (PCs) and metallothioneins (MTs). MTs are low-molecular-weight (7–8 kDa) polypeptides, rich in CC, CXC and CXXC motifs, that have been found in all kingdoms of life. MTs, in plants, are considered multifunctional proteins involved in essential-metal homeostasis. However, they can participate in the protection against HM toxicity by (i) the direct sequestration of HMs, particularly Cu(I), Zn (II) and Cd(II), (ii) scavenging reactive oxygen species (ROS) [137,138] and (iii) by regulating metallo-enzymes and transcription factors [139]. MTs are constitutively expressed but they are also induced by a wide variety of endogenous and exogenous stimuli and are temporally and spatially regulated [140]. In general, different types of MTs correlated with specific patterns of expression (spatial and temporal) (review in 140).

PCs are enzymatically synthesized peptides that are involved in HM binding [141]. PCs only contain three amino acids, glutamine, cysteine and glycine (Figure 3), and have been identified in many plant species and yeasts [142]. The first step of PC biosynthesis is catalysed by PC synthase and its starting compound is GSH. PCs are synthesized after exposure to HMs and are synthesized at different levels, depending on the specific HM; i.e., Cd and Pb induce higher levels of PCs than As and Cu [142]. PCs binds HM through the thiol group of cysteine, but the polymerization of PCs plays a role in the binding stability of the metal-PCn complexes [143]. The PC-metal complexes are transported from root to shoot or from shoot to root and, probably, through phloem [144].

Within the cells, organic acids, such as citrate and malate, the amino-acid derivative nicotianamine and phytate can also bind HMs, conferring heavy metal resistance to plants (reviewed in [145]). Outside the cells, organic acids and amino acids, such as citric and oxalic acids and histidine that are exudated by the plant, are also considered chelators of HMs, protecting plants from excessed of these ions [146,147]. The final step of heavy metal detoxification involves the sequestering of either free or chelated HMs into cell vacuoles. Finally, this PC-metal complexes are sequestered in vacuoles by specialized transporters ([148,149] and reviewed in [49]).

## 6. PAHs and HMs Produce Oxidative Stress in Plants

Plant PAH transformation enzymes, such as cytochrome CYP450, involve reduction or oxidation reactions that increase the levels of oxidants and harmful metabolites and activate the production of ROS [117]. The exposure of plants to HMs also elicits oxidative stress through two different mechanisms that depend on the different chemical properties of the metals [150]; (i) redox-active metals, under physiological conditions, exist in different oxidation states (i.e., Cu^+^/Cu^2+^ and Fe^2+^/Fe^3+^); this enables both metals to directly participate in the Fenton and Haber–Weiss reactions, leading to the formation of highly toxic hydroxyl radicals from H_2_O_2_ (Figure 4); (ii) physiologically non-redox-active metals, such as Cd, Hg and Zn, contribute only indirectly to increased ROS production, for example, by depleting or inhibiting cellular antioxidants (reviewed in [150]). Various enzymatic systems have been proposed to generate ROS in plants. These include a membrane-bound NADPH oxidase (similar to those found in neutrophils), lipoxygenase and apoplastic peroxidases [151].

When ROS production exceeds the antioxidizing capacity of the plant, the response can lead to cell death due to ROS toxicity and/or specific ROS-activated cell-death-inducing signalling events [152]. In *A. thaliana*, after exposure to atmospheric PAHs, a significantly increased production of reactive oxidative species (ROS) was observed, with concomitant necrosis of plant tissues and, therefore, inhibition of plant growth [132]. In wheat plants, microscopy studies revealed that cell structures become plasmolysed and distorted, and organelles disappeared as a consequence of the accumulation of H_2_O_2_ in plant tissues in response to the presence of 0.5 mg/L of phenanthrene [153]. The necrotic lesions produced by PAHs or HMs are similar to those produced in response to an avirulent pathogen in the hypersensitive response (HR) [154]. HR is characterized by the fast production and accumulation of ROS, primarily superoxide anions (O_2_^−^), hydrogen peroxide (H_2_O_2_) and the hydroperoxyl radical HO_2_, with the concomitant induction of local cell death to restrict the spread of the pathogen [154].

The ROS toxic effect within cells is exerted via lipid peroxidation, protein degradation modification and DNA damage [154] (Figure 4).

The most damaging consequence of ROS generation and accumulation is lipid peroxidation on cell and organelle membranes; in turn, the free fatty acid hydroperoxides can also be substrates of Fenton-like reactions, leading to the production of alkoxy radicals that enhance lipid peroxidation [155,156]. As a consequence, membrane fluidity increases with the concomitant cytosolic solute efflux and loss of functionality of membrane-associated proteins [157]. Furthermore, lipid peroxidation could result in the production of highly reactive aldehydes (i.e., malondialdehyde or 4-hydroxy-2-nonenal) that attack amino-acid side chains in proteins, causing protein damage and DNA fragmentation [158].

ROS-mediated post-translational modifications in proteins include sulphonylation, carbonylation, glutathionylation and s-nitrosylation [159], which are modifications that provoke protein malfunctioning, leading to cellular damage. H_2_O_2_ has been shown to hydroxylate cysteinyl thiols to form sulphenic acids. This oxidation is important in the formation of inter- and intramolecular disulphide bonds, as well as in the formation of disulphides with glutathione. These disulphides can be reduced to the thiol level through the activity of glutaredoxins or thioredoxins, with thiol oxidation being an important node for redox homeostasis [160]. Sulphonylation has been directly linked to the regulation of signalling and metabolic processes [161]; amongst the toxicological targets of oxidant stress induced by environmental contaminants are cysteinyl thiolate residues on many regulatory proteins [162]. S-glutathionylation is the subsequent modification of proteins; the sulphenic acid-containing side chains of proteins form covalent bonds with low-molecular-weight thiols, mainly with glutathione. This glutathionylation regulates the redox-driven signal transduction cascades and metabolic pathways [163] and can be reversed through thiol–disulphide oxidoreductase (thioltransferase) activity [164]. Protein carbonylation occurs in arginine, histidine, lysine, proline and threonine residues and it is considered an irreversible process [165]. The carbonylation of proteins can also be produced through indirect reactions of lipoperoxidation products with cysteine and histidine residues [166]. S-nitrosylation consists of the covalent binding of nitric oxide to thiol groups of cysteine residues, and it has been shown to modulate the signalling cascades of senescence, resistance and defence mechanisms [167]. S-nitrosylation has been involved in the modification of enzymes involved in respiration, antioxidation and photorespiration and it has also been reported to affect the DNA binding activity of some transcription factors [168,169].

The third main target of ROS accumulation in living cells are the electron-rich DNA bases; hydroxyl radicals attack the double bonds of the DNA bases producing di-, mono-, hydroxy-, and hydroxyl radicals, ring-saturated glycol, dehydrated, deaminated or ring-opened derivatives that further react to form stable DNA lesions, producing a diverse range of genotoxic modifications. As mentioned before, DNA bases may also be indirectly damaged through reaction with the products of lipid peroxidation, such as malondialdehyde, acrolein and crotonaldehyde. DNA sugars could also be damaged by ROS, leading to single-strand breaks. These lesions can be lethal, as they stop DNA replication, or by causing mutagenic changes in the replicated base [170].

To summarize, excessive production of ROS and subsequent oxidative damage is a common mechanism of phytotoxicity induced by both HMs and PAHs in plants. Independent, additive, synergistic and antagonistic toxic effects toward plants have been reported when plants were subjected to the combined pollution of PAHs and HMs [171,172,173,174]. However, to date, the mechanisms behind this synergistic or antagonistic toxicity of HMs and PAHs to plants is not fully understood [175]. HMs may induce damage to root cell membranes and consequently promote root uptake and the subsequent translocation of PAHs, thus increasing the damaging effects. On the other hand, HMs may cause lipid peroxidation of cell membranes and consequently decrease root lipid content, thereby decreasing the plant uptake of PAHs [176].

## 7. Plant Detoxification of Oxidative Stress Produced by PAHs and HMs

Plants respond to oxidative damage through the activation of the antioxidant machinery that triggers signalling cascades for stress tolerance. ROS antioxidant defence systems can be enzymatic and non-enzymatic, and both interact to neutralize free radicals. Proteomic studies have revealed that, in the presence of HMs and PAHs plants significantly increase the expression of superoxide dismutase, catalases, mono-dehydro-ascorbate reductase, ascorbate peroxidase, peroxiredoxins, glutathione-S-transferases, glutathione reductase, glutathione peroxidase and heat-shock proteins [53,177,178,179,180]. Enzymatic detoxification of ROS (Figure 5A) starts by the action of superoxide dismutase that converts the O_2_^−^ generated by NADPH oxidases into H_2_O_2_. The subsequent scavenging of H_2_O_2_ is carried out by catalases, ascorbate peroxidase, glutathione peroxidase, guaiacol peroxidase, class III peroxidases and peroxiredoxins. In general, peroxidases oxidize a wide variety of substrates, including H_2_O_2_ [181]. Catalases convert H_2_O_2_ to H_2_O and O_2_ without the use of reducing equivalents. Catalases have a high reaction rate but lower affinity of H_2_O_2_ than ascorbate peroxidases and, therefore, it has been suggested that catalases play a more important role in H_2_O_2_ detoxification than in the fine regulation of H_2_O_2_ as a signalling molecule [150].

Ascorbate, carotenoids, glutathione, polyamines, proline and α-tocopherol have been described as non-enzymatic antioxidants that also form part of the antioxidative defence system of plants [150,159] (Figure 5A). Ascorbate directly scavenge O_2_^−^, H_2_O_2_, and ^•^OH radicals and it is involved in the regeneration of other antioxidants [182]. Furthermore, it plays an important role in the ascorbate-glutathione cycle (Figure 5B). In this cycle, ascorbate peroxidase catalyses the conversion of H_2_O_2_ to H_2_O using ascorbate as the reducing agent. The reconversion of ascorbate to its reduced form is coupled to the oxidation of glutathione, which is subsequently reduced by the action of glutathione reductase [183].

As mentioned above, glutathione can also detoxify ROS [158,184], and plays an important role in the scavenging of metals [136,185].

α-tocopherol and carotenoids are important antioxidative metabolites involved in the protection against membrane lipid peroxidation and in the prevention of photosynthetic machinery damage, respectively [182]. Proline, that is accumulated in plants under many different types of abiotic stress, including HM exposure, is able to protect and stabilize ROS scavenging enzymes such as catalase and peroxidases [186]. In the presence of PAHs and HMs, plants also increase the synthesis of polyamides that can function as antioxidants, by conjugation to oxidative molecules and metals. It has also been suggested that polyamides can activate other cellular antioxidant defences, such as increased superoxide dismutase and enzymes associated with the ascorbate-glutathione cycle activities [187].

Plant phenolic compounds (such as coumarins, lignins, flavonoids, phenolic acids, or tannins) can remove ROS and chelate HMs by hydroxyl (-OH) and carboxylic acid (-COOH) [188,189,190]. The electron-donating, deprotonation equilibrium and radical-scavenging activity of phenolic compounds depends on their chemical structure, type, position and number of functional groups [191].

Whilst the participation of the majority of these mechanisms have been reported as the processes involved in responses toward the presence of HMs, carotenoid and superoxide dismutase seem to be the key factors for scavenging ROS in oxidative stress caused by PAHs in plant tissues [153].

## 8. Phytohormone Signalling Cascades in Plants in Response to PAHs and HMs

ROS are considered as signalling molecules that regulate plant development, biotic and abiotic stress responses [192]. Under normal conditions, ROS production is fine tuned to produce the appropriate physiological responses (for signalling, and metabolic processes). ROS responses depend on duration, site and concentration; the concentration and longevity of the ROS are determined by the composition and availability of antioxidant systems in each particular sub-cellular compartment [193]. Therefore, the rate of ROS diffusion and reactivity and ROS removal and perception, in the different cellular compartments of the plant, are highly regulated to create the so-called ROS network [192]. The fine equilibrium between ROS production and scavenging may be altered by different stresses. Low concentration of ROS acts as a signal (second messenger) and provokes a plant stress response; high ROS concentration causes cell damage and programmed cell death [194].

ROS are detected by ROS receptors. For example, the KEAP1 and NRF2 complexes are responsible for synchronizing plant stress responses in order to cope with various environmental and xenobiotic compounds. These stress signals are perceived and transmitted by histidine kinases, redox-sensitive transcription factors, ROS-sensitive phosphatases and redox-regulated ion channels [195]. All these systems activate signalling cascades that finally target the responsive genes, allowing plants to respond to many different environmental cues [195,196,197,198]. ROS production can directly alter the redox status of several enzymes and control metabolic fluxes in the cell [199]. It can also affect transcription and/or translation levels by modifying the function of some regulatory proteins (via ROS-derived redox modifications). These modifications can activate an adaptation response that would alleviate the effects of stress on cellular metabolism and reduce the level of produced ROS [199] or may also produce the so called “oxidative burst” that eventually leads to cell death [200,201,202,203,204].

ROS and heavy metals have been involved in the induction of mitogen-activated protein-kinase (MAPK) in alfalfa, rice (*Oryza sativa*) and *A. thaliana* [203,204,205,206,207]. The metal responsive transcription factor 1 (MTF-1) plays a significant role in the cellular response to heavy metal stress; this regulatory protein induces certain genes involved in heavy metal uptake and accumulation and ROS detoxification [208,209]. Proteomic studies have shown that the nucleoside, diphosphate kinase 3, is upregulated in plants exposed to PAHs; this kinase has a role in the metabolic and stress signalling functions and positively regulates enzymes involved in ROS detoxification such as catalases, ascorbate peroxidases, peroxiredoxins, glutathione-S-transferase and glutathione reductase [179]. Transcriptomic studies have revealed that the presence of PAHs, in addition to provoking alteration in the detoxification pathways of these molecules and ROS detoxification, also triggers signalling responses similar to pathogen defence mechanisms, including HR-like cell death and the induction of defence genes [210]. Phenanthrene-exposed plants showed induction of the expression of the pathogenesis related protein 1 (PR-1), a marker for HR and the glutathione-S-transferase gene GSTF2, which is induced by ethylene, auxin, salicylate, paraquat and several sulfhydryl compounds [132]. This suggests that phytohormones are also produced in response to PAHs (Figure 6).

Phytohormones are plant-endogenous molecules that modify physiological and molecular reactions in response to different cues and are critically required for plant survival under abiotic stress [211]. Therefore, it has been amply demonstrated that the accumulation of ROS affects the level and function of many plant hormones, including ethylene [212,213], abscisic acid (ABA), gibberellic acid (GA), brassinosteroids, NO and phytohormone-modulating stress response reactions, such as salicylic acid (SA) and jasmonic acid (JA), and development-associated hormones, such as auxins and cytokinins [132,212,214,215,216,217,218].

There is a great deal of evidence to support the induction of genes regulated by phytohormones in response to environmental contaminants; genes encoding the ethylene-inducible defence response proteins, PDF1.2a and PDF1.2b, are strongly upregulated in *A. thaliana* in response to cadmium [219,220]; the pathogenesis-related gene, *PR-1*, a marker gene for systemic acquired resistance and HR responses and regulated by SA, is highly upregulated in PAH-exposed plants. Although ethylene-, JA- and SA-mediated responses are induced by PAHs, the induction of *PR-1* does not require the production of ethylene or jasmonate and, therefore, it has been suggested that PAHs independently induce both signalling pathways [210].

The presence of HMs also activates a complex signalling network, wherein phytohormones and ROS can play complementary or an antagonistic roles [221]. Exposure to HMs induces the endogenous levels of ABA, auxins, brassinosteroids, ethylene, GAs, JAs and SA [211,222,223,224,225,226,227] and reduces the levels of cytokinins [228]. ABA transcriptionally regulates up to 10% of protein-encoding genes in *Arabidopsis* [229,230]. Although the mechanism of ABA in response to HMs is not well known, it has been suggested that it might regulate stomata closure to regulate water balance in plants under cadmium stress [231]. The elevated levels of indole-3-acetic acid (IAA) have been connected with plant growth reduction, which can be a result of hormonal unbalance under stress conditions [211]. Brassinosteroids are plant steroids involved in the regulation of the anti-oxidative system of plants and help to support plant growth under heavy metals stress [232]. It has been described that some of them have the potential to directly reduce heavy metals, diminishing their deleterious effects [225]. GAs positively affects seed germination, stem elongation, leaf expansion, flower and trichome initiation and the development of fruits and supports plant adaptation and resistance to abiotic stress among them, protection against the toxic effects of HMs [221]. JA, and its derivatives, protect plants from the toxic effects of HMs by enhancing the production of non-enzymatic antioxidants, such as phenolic compounds and enzymatic antioxidants such as superoxide dismutase and balance the production of photosynthetic pigments [226]. Under normal conditions, SA is a major regulator of photosynthesis influencing chlorophyll content, stomatal conductivity, and photosynthesis-related enzyme activity in plants [227].

Interactions amongst different hormones have also been described during HM stress. For example, ethylene modulates root morphogenesis during HM stress in *A. thaliana* by increasing the production of auxins and the activity of superoxide dismutase (SOD) isoenzymes responsible for the control over superoxide accumulation [224]. Cytokinins, which under normal conditions play a regulatory role in modulating plant development [228], have been described as antagonists of ABA and modifications in the levels of both plant hormones under HM stress can be dependent on each other as a result of their crosstalk [221]. SA, under heavy metal stress, also interacts with other plant hormones (such as GAs, auxins, or ABA) promoting the stimulation of the production of antioxidant compounds and enzymes. These interactions have been described as an alerting system in HM-stressed plants, helping them to cope with HM stress [233]. Signalling networks produced by ROS and its cross-talk with HMs have been widely reported in plants but less so for PAHs. However, the activation of the production of phytohormones under PAH and HM stress suggests parallelisms between the pathogen-elicited responses and the responses toward contaminants.

The upregulation of some auxin-related genes in the presence of the LMW-PAH naphthalene has been explained by the structural similarities of this compound with the plant growth regulator naphthalene acetic acid. In such a way, not only ROS responses, but also the absorption of the contaminant, could trigger the responses that may help plants to cope with pollutant stress [118].

miRNAs, although less studied, also play an important role in the signalling of heavy metal stress. miRNAs are a class of 21–24 nucleotide non-coding RNAs involved in post-transcriptional gene silencing by their near-perfect pairing with a target gene mRNA [234]. Sixty-nine miRNAs were induced in *Brassica juncea* in response to arsenic; some of them were involved in regulation of indole-3 acetic acid, indole-3- butyric and naphthalene acetic acid, JAs (jasmonic acid and methyl jasmonate) and ABA. Others were regulating sulphur uptake, transport and assimilation [235].

Phytohormone alterations lead to metabolic modifications; i.e., in the presence of PAHs, plant tissues are able to overproduce osmolytes such as proline, hydroxyproline, glucose, fructose and sucrose [236]. Proline biosynthesis and accumulation is stimulated in many plant species in response to diverse environmental stresses (such as water deficit, and salinity) triggered by factors such as salicylic acid or ROS [186]. The overproduction of hydroxyproline, which could be explained by the reaction between proline and hydroxyl radicals [237], and of sucrose have also been observed [238,239]. This accumulation of osmolytes also seems to be regulated by ABA, whose levels are increased in plants exposed to PAHs [210].

## 9. Conclusions and Future Perspectives

Pollutants induced a wide variety of responses in plants leading to tolerance or toxicity. The myriad of plant responses, responsible for the detection, transport and detoxification of xenobiotics, have been defined as xenomic responses [240]. The emergence of –omic techniques has allowed the identification of many of these responses, although these types of studies are still too scarce to be able to draw a definitive map of the plant pathways that cope with pollutant stresses. Many of the plant responses are common to those observed with other stresses (i.e., production of ROS), however, some others do seem to be specific (transport and accumulation in vacuoles or cell walls). The identification of HM and PAH plant receptors and the subsequent specific signal cascades for the induction of specific responses (i.e., the synthesis of phytochelatins or metallothioneins) are aspects that remain to be explored.

The holobiont, the supraorganism which the plant produces with its associated microbiota, also has relevance in the context of plant responses toward contaminants. Whilst the mechanisms by which plants can activate the metabolism of the microbiota, or the specific selection of microbial genotypes that favour plant growth, have been explored for more than two decades, these aspects have been less studied under contaminated environments and have mainly been studied in soil systems [38,121,241,242]. The presence of PAHs increases the “nutrient” content for some of the associated microorganisms, provoking changes in the microbial composition and metabolism [243,244]. How these alterations influence the capacity of the plant to respond to contaminants is an unexplored question. There are some studies dealing with the changes in the phyllospheric microbiota composition in response to atmospheric pollution [245,246], however, there are very few studies dealing with the specific plant-microbe interactions in the contaminated phyllosphere. Aspects, such as how plants can cope with the intermediates of PAH degradation and the effects that the presence of these intermediates in roots or leaves can exert over plant physiology, have been the subject of much study. One of the main targets for these studies is salicylic acid, which is an intermediate of the PAH degradation as well as plant hormones.

How plants modulate and coordinate all these responses should drive the improvements in the utilization of these responses in phytoremediation. Furthermore, it can be assumed that low levels of contaminants may lead to a basal resistance toward other biotic or abiotic stresses, therefore, an open question is whether the stimulation of the defensive system of plant by low quantities of contaminants could became an acceptable technique for crop protection. Another open question is the possibility of a commercial production of added-value compounds during plant growth under pollutant-derived stress. It has been suggested that the presence of heavy metals may serve to stimulate the production of bioactive compounds with pharmaceutically important properties [247]. For example, α-linolenic acid, which increases during HM exposure in plants, is a precursor of long-chain n-3 polyunsaturated fatty acids, such as eicosapentaenoic acid and docosahexaenoic acid, which have important applications as anti-inflammatory, anti-thrombotic and anti-neurodegenerative medication [57]. Other compounds that increase during HM exposure in plants are saponins (that have pharmaceutical as well industrial interest as food additives or the components of photographic emulsions), cyclic hydroxamic acids (as insecticides, antimicrobials, anti-malarials and others) and sesquiterpenes, and isoflavonoids and sulphur-containing compounds, which are potential antioxidants [67,248].

Therefore, although plant responses toward pollutants are similar to responses to other stresses, and many have been extensively studied (such as the production of ROS), there are still many open questions regarding how plants sense contamination and how they are able to modulate their responses. The tolerance/sensitivity of plants is mediated by many different processes that have to be coordinated for survival; the mechanisms by which cause this cross-regulation to happen are still unknown. Finally, how these processes could be improved for bioremediation or for industrial processes is an interesting and open field of research.

## Figures and Tables

**Figure 1 plants-10-02305-f001:**
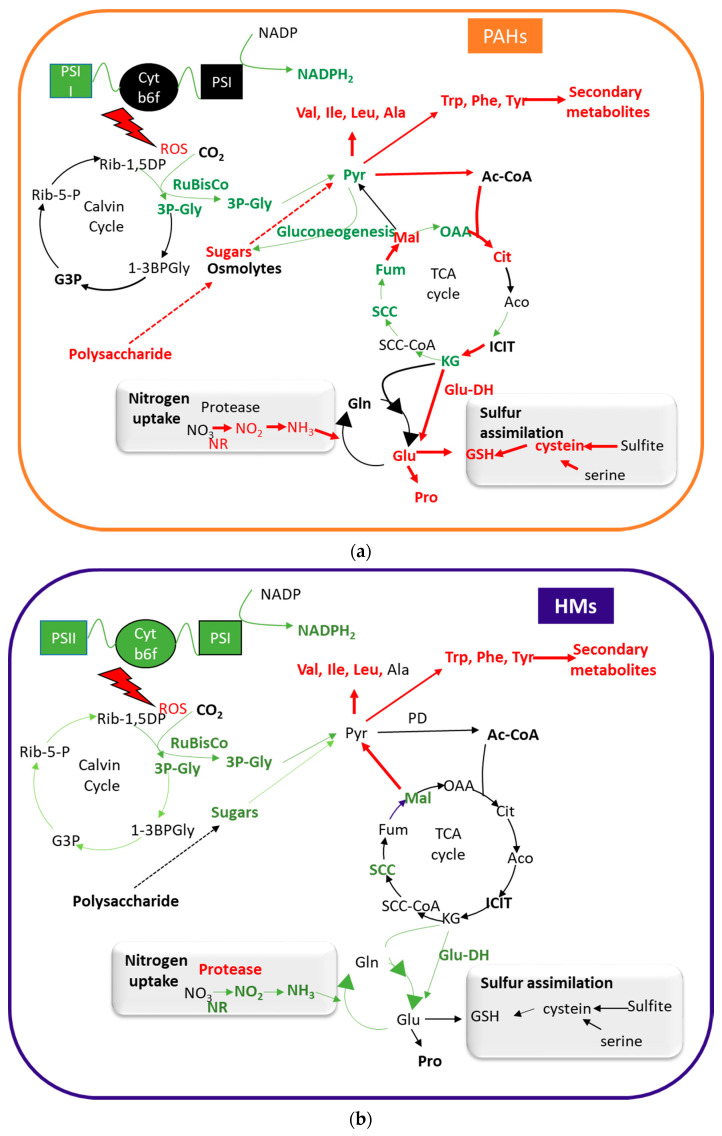
Schematic representation of the metabolic processes that have been reported as induced or repressed in the presence of PAHs (**a**) and HMs (**b**). Red indicates the processes (enzymes and compounds) that are induced in the presence of the contaminants, and green those that are repressed. NR: nitrate reductase; Glu-DH: glutamate dehydrogenase; PD: pyruvate dehydrogenase complex; GSH: glutathione; PSI: phosphosystem I; PSI: phosphosystem II; Cytb6f: cytochrome b6f; Pyr: pyruvate; Ac-CoA: acetyl-CoA; Cit: citrate; Aco: aconitate; ICIT: isocitrate; KG: α-ketoglutarate; SCC-CoA: succinyl-CoA; SCC: succinate; Fum: fumarate; Mal: malate; OAA: oxaloacetate; 3P-Gly: 3-phosphoglycerate; G3P: glyceraldehyde-3-phosphate; Rib-5-P: ribulose 5-phosphate; Rib-1,5DP: ribulose-1,5-bisphosphate; 1-3 BPGly: 1,3-bisphosphoglycerate.

**Figure 2 plants-10-02305-f002:**
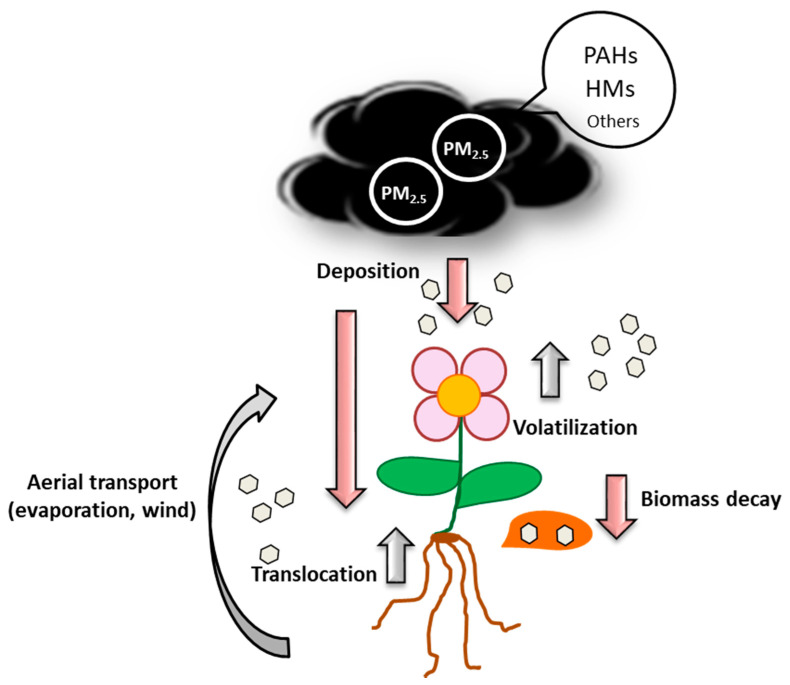
Schematic representation of the processes involved in the air–soil–plant mobilization of PMs (modified from [78]).

**Figure 3 plants-10-02305-f003:**
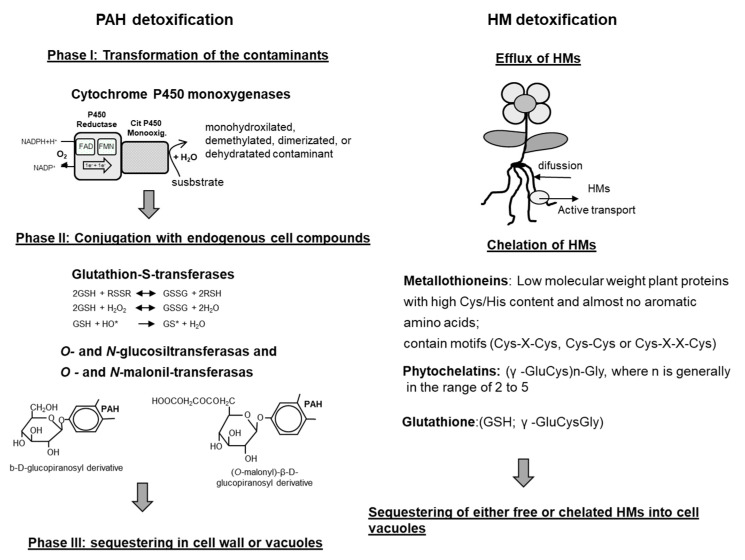
Main plant activities and elements involved in the detoxification of PAHs and HMs.

**Figure 4 plants-10-02305-f004:**
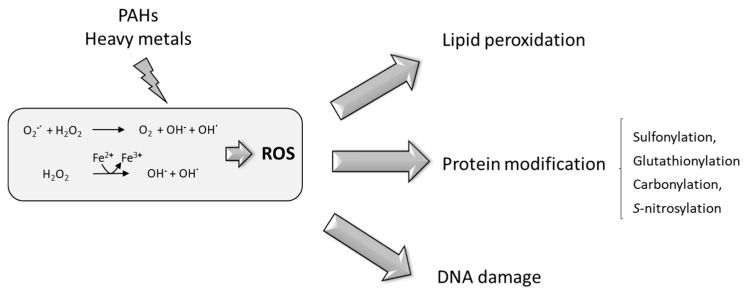
Schematic representation depicting the Haber-Weiss and Fenton reactions leading to ROS and its effects on lipids, proteins and DNA.

**Figure 5 plants-10-02305-f005:**
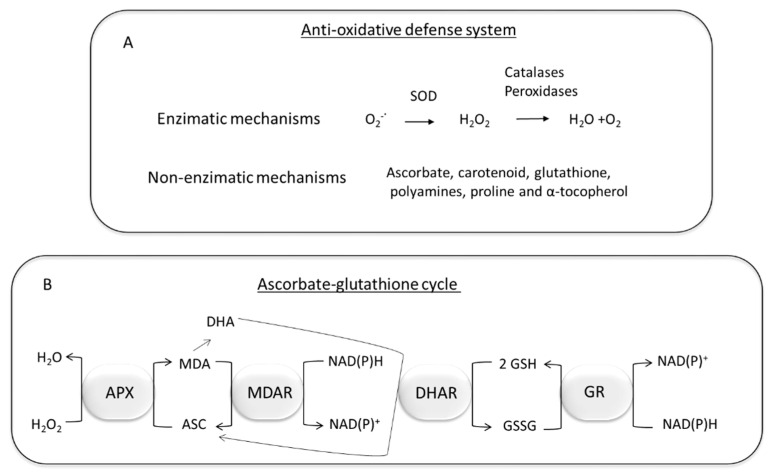
Schematic representation of the anti-oxidative defence system in plants (**A**) and the ascorbate-glutathione cycle. (**B**) SOD: Superoxide dismutase; APX: ascorbate peroxidase; ASC: ascorbate; GSH glutathione; MDA: monodehydroascorbate; MDAR: monodehydroascorbate reductase; DHA: dehydroascorbate; DHAR: dehydroascorbate reductase; GSSG: oxidized glutathione; GR glutathione reductase (modified from [183]).

**Figure 6 plants-10-02305-f006:**
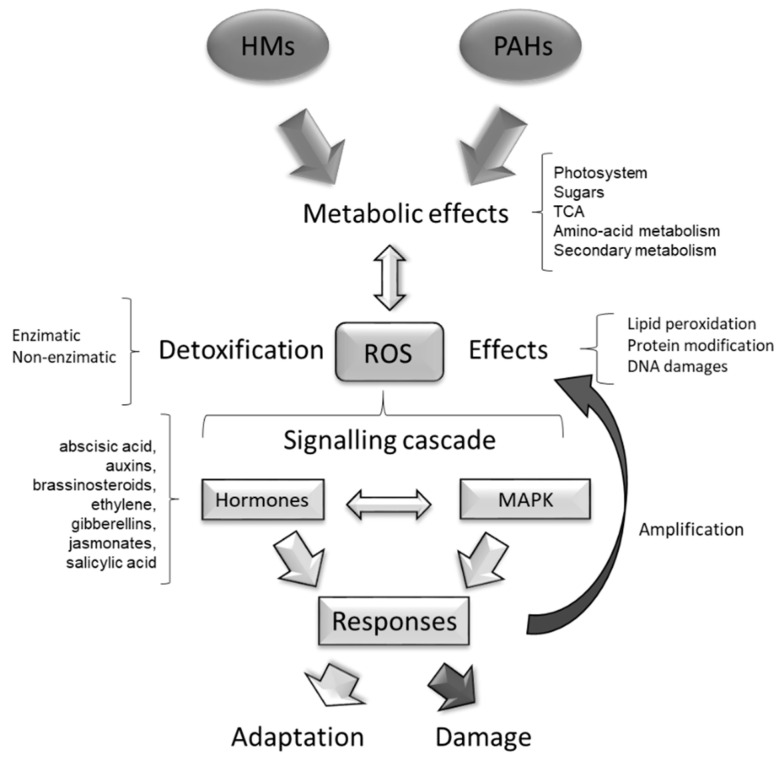
Schematic representation of the cascade of responses triggered by HMs and PAHs in plants. Depending on the intensity of the process, adaptation to stress or cellular damage and cell death is the final outcome of the process.

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
