# Peer review of "Biochemical and Metabolic Plant Responses toward Polycyclic Aromatic Hydrocarbons and Heavy Metals Present in Atmospheric Pollution"

_plants, 2021, doi:10.3390/plants10112305_

Round 1
Reviewer 1 Report
The work presented by the authors entitled Plant responses to atmospheric pollution is an extensive overview of the effects of PAH and trace metals on plants. The subject of the article is very interesting, the manuscript reads very well, it has a logical layout, division into individual chapters and subsections makes it easier for the reader to navigate in the text. However, I have some comments and suggestions.
The authors decided to compare the impact and fate in the plant of two extremely important abiotic stress factors. Due to the wide range of topics, some aspects are described quite vaguely or in a simplified way, especially in relation to trace metals.
p 2 w 54 I think the authors should look again at source [9] and slightly modify the sentence regarding the sources of HM in the atmosphere. As in the case of PAH, it is also possible to take into account natural sources such as weathering, volcanic eruptions or microbiological activity, as well as various anthopogenic sources such as burning fossil fuels, production of glass, fertilizers, semiconductors, waste incineration, plastics production and many other branches of industry in addition to mining and metallurgy
p 3 w 126 I do not understand the reference, for PAH the authors wrote about a low level inducing the effect of hormase and for HM the sublethal dose is similar? Additionally, I think that the authors greatly simplified the reference to the source [43], where several hypotheses related to the hormesis effect were presented. In the literature, the hormesis effect is most often shown for plants treated with a low level of heavy metals .
s 3 w 147 in the sentence sa two "directly"
p 5 w 215 The authors equate metallophytes with hyperaccumulators, which again is a huge simplification. If, as for PAH, the authors limit the description to only hyperaccumulators, the beginning of the paragraph should be modified.
p 7-8 section Detoxification of HM - again, the authors presented the fate of HM in a simplified manner compared to the detoxification process described for PAH. There are many organic and inorganic ligands in the apoplastic space and in cells that are involved in metal binding. Additionally, the individual ligands have both species and ion specificity, such as, for example, PC against Cd in tobacco and MT against Cu in Selene vulgaris. Therefore, I would suggest extending this subsection.
p 11 w 449-453 The information in this paragraph is already mentioned in the manuscript, hence I would suggest deleting it.
s 14 fig 6 Unfortunately, I find the drawing illegible, the light green font is very difficult to read, only after a large magnification I was able to recognize all the elements.
Throughout the manuscript, the authors use common plant names, but in several places they use Latin names or both - I think the authors should take one way - it seems the authors preference is to use common names, but they should give the correct species name upon first use.
I also have a general remark, in my opinion chapter 8 related to the effects of PAH and HM on metabolism should be placed as chapter 3 - although of course this is my personal preference and the authors do not have to be guided by it.
Author Response
Reviewer 1
- The authors decided to compare the impact and fate in the plant of two extremely important abiotic stress factors. Due to the wide range of topics, some aspects are described quite vaguely or in a simplified way, especially in relation to trace metals.
We agree with the reviewer’s comment. Our manuscript covers many different aspects of the impact of two widespread contaminants in plants, and therefore, we have simplified some aspects and we have referred to the corresponding extensive reviews in the bibliography to cover some specific aspects related with those topics. However, we agree that, probably because we are mainly working with PAHs, there are some aspects about metals that have been poorly described and therefore, we have tried to improve these parts of the manuscript.
- p 2 w 54 I think the authors should look again at source [9] and slightly modify the sentence regarding the sources of HM in the atmosphere. As in the case of PAH, it is also possible to take into account natural sources such as weathering, volcanic eruptions or microbiological activity, as well as various anthopogenic sources such as burning fossil fuels, production of glass, fertilizers, semiconductors, waste incineration, plastics production and many other branches of industry in addition to mining and metallurgy.
We have modified the sentence and the paragraph including other sources of HM contamination and we have explained how human activities have modified the local concentrations or location of these elements.
- p 3 w 126 I do not understand the reference, for PAH the authors wrote about a low level inducing the effect of hormase and for HM the sublethal dose is similar? Additionally, I think that the authors greatly simplified the reference to the source [43], where several hypotheses related to the hormesis effect were presented. In the literature, the hormesis effect is most often shown for plants treated with a low level of heavy metals.
The reviewer is right; after re-reading this part of the MS we realized that maybe the term hormesis was not well defined in the text and the meaning of the paragraph was not clear enough. We have now defined the term and we have written the general effect of hormesis in HMs and PAHs.
- s 3 w 147 in the sentence sa two "directly"
We have deleted one of the directly.
- p 5 w 215 The authors equate metallophytes with hyperaccumulators, which again is a huge simplification. If, as for PAH, the authors limit the description to only hyperaccumulators, the beginning of the paragraph should be modified.
The reviewer is absolutely right; we have now modified the paragraph; we believe that now it is clear that only some metallophytes are consider as hyperaccumulators.
- p 7-8 section Detoxification of HM - again, the authors presented the fate of HM in a simplified manner compared to the detoxification process described for PAH. There are many organic and inorganic ligands in the apoplastic space and in cells that are involved in metal binding. Additionally, the individual ligands have both species and ion specificity, such as, for example, PC against Cd in tobacco and MT against Cu in Selene vulgaris. Therefore, I would suggest extending this subsection.
We have extended this sub-section; however, the information about PCs and MTs and their role in metal homostasis have been extensively reviewed and it is impossible to summarize all this information in this review. As mentioned above, we have included many reviews dealing with different aspects of these part of the review in the bibliography.
- p 11 w 449-453 The information in this paragraph is already mentioned in the manuscript, hence I would suggest deleting it.
We have shortened this paragraph following the suggestion of the reviewer.
- s 14 fig 6 Unfortunately, I find the drawing illegible, the light green font is very difficult to read, only after a large magnification I was able to recognize all the elements.
We have modified the figure according to the reviewer´s suggestions.
- Throughout the manuscript, the authors use common plant names, but in several places they use Latin names or both - I think the authors should take one way - it seems the authors preference is to use common names, but they should give the correct species name upon first use.
We thank the reviewer for letting us know about this inconsistency. We opted for the common names of well-known plants (especially for crops) for making the text easier for the broader audience. Latin names are used for less well known plants. However, we have now indicated the scientific name the first time we use it.
- I also have a general remark, in my opinion chapter 8 related to the effects of PAH and HM on metabolism should be placed as chapter 3 - although of course this is my personal preference and the authors do not have to be guided by it.
The reviewer is absolutely right; it makes sense to move this part of the manuscript to after the effects on seed germination and plant growth.
Reviewer 2 Report
I think this paper is a useful and timely review and synthesis. I found the writing style easy to follow with only minor corrections to typographical errors needed. Although I realize that it is not the focus of the review, I do think some of the research/papers regarding TCE remediation through transpiration streams should be incorporated and briefly mentioned.

Author Response
Review 2
I think this paper is a useful and timely review and synthesis. I found the writing style easy to follow with only minor corrections to typographical errors needed. Although I realize that it is not the focus of the review, I do think some of the research/papers regarding TCE remediation through transpiration streams should be incorporated and briefly mentioned.
We thank the reviewer for their comments. In fact, TCE is an important contaminant, and its uptake by plants has been studied in detail. We did not include this contaminant in the review because it is mainly considered as a soil and groundwater contaminant, although it is also present in the air. There are other important contaminants that have also not been reviewed (PCB, BTEX and others) to limit the scope of the review. However, and as suggested by reviewer 3, we have modified the title of the manuscript to clearly determine its scope.
Reviewer 3 Report
The analyzed review is extensive, based on a more than generous bibliography. The authors make a consistent synthesis on a topical issue, for which there is a lot of literature - the impact of air pollution on plants.
The paper is well organized, the chapters have a logical sequence and the thread is easy to follow.
The authors can reanalyze the title: it is a bit comprehensive, the information in the text refers mainly to metabolic and biochemical reactions of plants, other aspects - such as anatomical or ultrastructural changes, widely analyzed in the literature, are not touched on paper.
I have only a few small remarks to make:
row 107 ”damage suffered by the presence ”may be ”damage suffered in the presence ”?
row 162 ”root suberine cortical zones” - maybe exodermis?
In Subchapter 3.3. Accumulation - you can also consider synthesis paper regarding the database for plants that hyperaccumulate metals here: https://nph.onlinelibrary.wiley.com/doi/10.1111/nph.14907
This paper was scanned with the Turnitin program and the result of the degree of similarity is 30% (see the attached document). As this is a review, it is normal for most of the information to be taken from the consulted scientific literature. The citation in the text is correct. However, I recommend revising some paragraphs (where there are 4-5 rows taken compactly from the cited paper), in order to reduce this similarity index.
Another suggestion is related to the illustration of the article - if the figures are original, it is good to specify this.
Overall, I liked the paper, I congratulate the authors for their work and I wish them much success!

Author Response
Review 3
We are very grateful for the reviewer comments; we are sure we have improved the manuscript by following their recommendations.
- The authors can reanalyze the title: it is a bit comprehensive, the information in the text refers mainly to metabolic and biochemical reactions of plants, other aspects - such as anatomical or ultrastructural changes, widely analyzed in the literature, are not touched on paper.
We have changed the title as suggested: “Biochemical and metabolic plant responses to atmospheric pollution”; it is true that there are other plant modifications in response to contaminants that are not mentioned in the text. The review was extensive enough without including other responses.
- row 107 ”damage suffered by the presence ”may be ”damage suffered in the presence ”?
We agree with the reviewer and, accordingly, we have changed the text.
- row 162 ”root suberine cortical zones” - maybe exodermis?
We have modified this part of the text; suberine in mainly located in the endodermis.
- In Subchapter 3.3. Accumulation - you can also consider synthesis paper regarding the database for plants that hyperaccumulate metals here: https://nph.onlinelibrary.wiley.com/doi/10.1111/nph.14907
Thanks for the tip; it is a very interesting paper. We have incorporated the reference.
- This paper was scanned with the Turnitin program and the result of the degree of similarity is 30% (see the attached document). As this is a review, it is normal for most of the information to be taken from the consulted scientific literature. The citation in the text is correct. However, I recommend revising some paragraphs (where there are 4-5 rows taken compactly from the cited paper), in order to reduce this similarity index.
Thanks a lot for letting us know about this problem. We have incorporated the necessary changes throughout the manuscript to correct it.
- Another suggestion is related to the illustration of the article - if the figures are original, it is good to specify this.
We have specified and referenced the figures that are based on published manuscripts.